# The Role of Antimicrobial Peptides in Preterm Birth

**DOI:** 10.3390/ijms22168905

**Published:** 2021-08-18

**Authors:** Ga-Hyun Son, Jae-Jun Lee, Youngmi Kim, Keun-Young Lee

**Affiliations:** 1Division of Maternal-Fetal Medicine, Department of Obstetrics and Gynecology, Hallym University College of Medicine, Kangnam Sacred Heart Hospital, Seoul 07441, Korea; mfmlee@hallym.ac.kr; 2Institute of New Frontier Research Team, College of Medicine, Hallym University, Chuncheon 24523, Korea; iloveu59@hallym.or.kr (J.-J.L.); kym8389@hanmail.net (Y.K.); 3Departments of Anesthesiology and Pain Medicine, College of Medicine, Hallym University, Chuncheon 24523, Korea

**Keywords:** antimicrobial peptides, preterm birth, defensin, innate immunity

## Abstract

Antimicrobial peptides (AMPs) are short cationic amphipathic peptides with a wide range of antimicrobial properties and play an important role in the maintenance of immune homeostasis by modulating immune responses in the reproductive tract. As intra-amniotic infection and microbial dysbiosis emerge as common causes of preterm births (PTBs), a better understanding of the AMPs involved in the development of PTB is essential. The altered expression of AMPs has been reported in PTB-related clinical presentations, such as preterm labor, intra-amniotic infection/inflammation, premature rupture of membranes, and cervical insufficiency. Moreover, it was previously reported that dysregulation of AMPs may affect the pregnancy prognosis. This review aims to describe the expression of AMPs associated with PTBs and to provide new perspectives on the role of AMPs in PTB.

## 1. Introduction

Preterm birth (PTB) refers to a multi-etiological condition that occurs in more than one out of ten child births, and approximately 1.1 million neonates die from prematurity-related complications each year [1]. Moreover, prematurity is a huge burden on the healthcare system because of long-term morbidities, such as neurodevelopmental disabilities and immediate complications related to organ system immaturity [2,3,4,5,6]. Approximately 70% of PTBs occur spontaneously due to preterm labor or preterm premature rupture of membranes (PPROM), whereas medically indicated PTBs are often preceded by maternal or fetal complications, including preeclampsia or intrauterine growth restriction [7,8,9]. Intra-amniotic infection is thought to contribute to at least one-third of spontaneous PTBs, and early gestational age is associated with a higher frequency of intra-amniotic infection [10,11,12,13,14]. The mechanism by which microorganisms enter the amniotic cavity is unclear. However, considering that microorganisms frequently found in intra-amniotic infection are common constituents of vaginal microbiome, intrauterine infection could often be a result of ascending infection by lower genital tract microorganisms [15,16,17].

The cellular structure of the reproductive tract and various immunological functions form a barrier against microbial pathogens during pregnancy. Uterine, vaginal, and cervical epithelial cells build a physical barrier by expressing intercellular junctions, including tight or occluding junctions and adherent junctions [18,19,20,21]. Furthermore, an array of immune cells is expressed in the reproductive tract to form a mucosal barrier, and mucosal epithelial and immune cells express pattern recognition receptors (PRRs) to sense and respond to pathogens [19,22,23,24,25,26,27]. In particular, the mucus plug in the cervical canal not only forms a physical barrier between the vagina (exposed to microbes) and the intrauterine space (considered “sterile”) during pregnancy, but also contains various immune cells, cytokines, chemokines, and antimicrobial peptides (AMPs), thereby serving as an important antimicrobial barrier [28,29]. AMPs are a class of small peptides with a wide range of antimicrobial properties and are expressed throughout the female reproductive tract, including the cervix, vagina, uterine wall, amniotic fluid, fetal membranes, placenta, and fetus [30,31,32]. Recently, AMPs have been found to perform immunoregulatory functions through diverse mechanisms and play an essential role in maintaining immune homeostasis by interacting with the microbiome [33,34]. Several recent studies have reported an association between alterations in AMP expression and release, and infectious and inflammatory diseases, such as Crohn’s disease and atopic dermatitis [35,36,37,38]. Further, recent studies have reported that alterations in AMP expression are associated with PTB. Intra-amniotic infection and inflammation are associated with higher concentrations of AMPs, and low cervicovaginal β-defensin levels in the mid-trimester were associated with a higher risk of spontaneous PTB [39,40,41].

Considering that intrauterine infection or inflammation and vaginal dysbiosis are some of the key triggers of PTB, AMPs may have an array of effects on PTB. Therefore, in this review, we highlight the recent findings regarding the altered expression of PTB-associated AMPs and their potential influences on the development of PTB.

## 2. AMPs in PTB

### 2.1. Properties and Classification of AMPs

AMPs are evolutionarily conserved molecules that are produced in all multicellular organisms from prokaryotes to humans, and they are the first-line defense against microbial pathogens [34,42,43,44]. AMPs have wide-ranging antimicrobial functions against bacteria, yeast, fungi, and viruses [33,44,45,46]. In higher eukaryotic organisms, AMPs also have diverse immunoregulatory activities, based on which they have also been called host defense peptides (HDPs) in recent years [34,47,48]. AMPs are comprised of 10–50 amino acids with varying positive charges (from +2 to +9) and are amphipathic peptides with basic amino acids on one end, and hydrophobic residues on the other end, thereby constituting a unique structure that is water-soluble, positively charged, and hydrophobic [49,50,51]. Currently, more than 2600 natural AMPs with diverse sequences and structures are known [52]. AMPs can be broadly classified into α-helical, β-sheet, and extended AMPs based on their secondary structure. Most AMPs belong to the α-helical and β-sheet categories [51,53,54,55]. The two main classes of AMPs are defensins and cathelicidins. Defensins have a common β-sheet core stabilized by three disulfide bonds between six conserved cysteine residues and are classified into α-, β-, and θ-defensins based on the configuration of the disulfide bonds [56,57]. Several human α-defensins are highly expressed in neutrophils; thus, they are known as human neutrophil peptides 1–4 (HNP1–4). HNP1–4 are stored in the azurophilic granules of neutrophils and account for 30–50% of the protein content in azurophilic granules [44,58]. Other α-defensins are produced by Paneth cells of the small intestine (human α-defensin 5 and 6; HD5 and HD6) [59]. Production and secretion of HNP1–3 can be upregulated by pro-inflammatory cytokines in immature monocyte-derived dendritic cells, whereas HNP1 and HD5 expressions depend on the nucleotide-binding oligomerization domain-containing protein2 (NOD2) stimulation in gut epithelial cells [55,60,61,62]. Human β-defensins (HBDs) are mainly produced in epithelial cells and have a protective function in sites that are exposed to microbes, such as the respiratory, intestinal, and genitourinary tracts and skin [55,63]. HBD-1 is constitutively expressed, whereas HBD-2 and HBD-3 are induced by a variety of inflammatory stimuli [63,64]. Cathelicidins, which are α-helical peptides, are generally produced by epithelial cells and many immune cells. Humans have only one cathelicidin gene (CAMP) [65,66]. Cathelicidins are synthesized as a prepropeptide known as hCAP18 and can be converted into several cathelicidine peptide variants through proteolytic cleavage by various proteases, one of the most common variants being LL-37 [67]. In epithelial cells, expression of LL-37 is modulated by inflammatory stimuli; however, vitamin D also influences LL-37 expression [65,66,68]. Vitamin D3 directly increases hCAP18 transcription and synergizes with lipopolysaccharide (LPS) in LL-37 production in neutrophils. In addition, vitamin D3 plays an important role in LL-37 induction through Toll-like receptor 1 (TLR1) and TLR2 pathways during bacterial infection in monocytes and keratinocytes [66,68].

### 2.2. Mechanism of Action of AMPs

#### 2.2.1. Antimicrobial Actions

AMPs have direct and rapid antimicrobial activities by destroying the physical integrity of microbial membranes or by acting on intracellular targets via membrane translocation [50]. Bacterial membranes are a major target of cationic AMPs, and because bacterial membranes are negatively charged due to anionic lipids, such as phosphatidylglycerol and phosphatidylserine, they electrostatically interact with positively charged AMPs [49]. In addition, teichoic acids in the cell walls of Gram-positive bacteria and LPS in the outer membranes of Gram-negative bacteria further impart a negative charge to bacterial membranes [69]. The key mechanism of the antimicrobial action of AMPs involves the formation of an amphipathic secondary structure upon contact with cytoplasmic membranes, which ultimately leads to membrane perturbation, bacterial cell content leakage, and cell death [70,71]. To explain membrane perturbation by AMP, models that disrupt membranes by forming barrel-stave and toroidal pores and directly destroy the membrane by thinning and dissolving the lipid bilayer (carpet model) have been proposed. However, these models are of limited utility, as they are based on experiments that used model membranes [54,71,72]. Unlike bacteria, mammalian cell membranes generally consist of zwitterionic phospholipids and thus have a neutral net charge, wherein phospholipids with negatively charged head groups face inward [49,73,74]. This results in a relatively weak hydrophobic interaction with AMPs, and the high cholesterol content in mammalian cell membranes reduces AMP activity by stabilizing the phospholipid bilayer [49,75]. In addition to the antimicrobial action through membrane perturbation described above, AMPs may also exhibit antimicrobial effects by acting on intracellular targets, such as nucleic acids and proteins, through membrane translocation. Although the exact mechanism of membrane translocation by AMPs remains unknown, the involvement of inner membrane transporters or transient pores has been proposed. In addition, AMPs exhibit antiviral activity by destabilizing the viral envelope and damaging the virions, or inhibiting the viral replication of non-enveloped viruses to prevent the nuclear entry of the viral genome [76,77,78].

#### 2.2.2. Immunoregulatory Functions of AMPs

Recent studies have demonstrated that AMPs perform a broad range of immunomodulatory functions beyond their antimicrobial activity [33]. The molecular mechanism by which AMPs regulate immune responses is highly complex, and immunomodulatory activity varies depending on the environmental stimuli, cell type, and peptide concentration [33,48]. Immunomodulatory activities of AMPs include stimulating chemotaxis of immune cells, modulating neutrophil function, and influencing adaptive immunity by recruiting antigen-presenting cells to infection sites. AMPs function as chemoattractants to stimulate chemotaxis of leukocytes by secreting chemokines [55,63,79,80]. In addition, AMPs regulate neutrophil functions by stimulating the release of neutrophil chemokines or increasing neutrophil influx through chemotactic functions [81,82]. AMPs are also present in neutrophil extracellular traps (NETs) and are involved in NET-mediated antibacterial effects [83]. Upon infection, AMPs recruit antigen-presenting cells, such as monocytes, macrophages, and dendritic cells, and mediate innate and adaptive immunity. AMPs exert both pro- and anti-inflammatory properties depending on the cell type and inflammatory stimuli, thereby establishing a balance of inflammation. The anti-inflammatory functions of AMPs are highlighted in studies on the association between low α-defensin expression and ileal Crohn’s disease, and in a report that a cathelicidin-knockout mouse model showed more severe inflammatory responses than the wild type [84,85,86,87]. On the other hand, cathelicidin can also promote inflammation through the induction of proinflammatory cytokines and chemokines or DNA- and RNA-mediated TLR activation [88,89,90]. Thus, overproduction of AMPs can directly trigger inflammatory diseases such as psoriasis, which highly express AMPs such as cathelicidin, β-defensins, and S100 proteins in their lesions [91].

#### 2.2.3. AMP-Microbiome Interaction

There is mounting recent evidence supporting that the commensal microbiota in the body plays a pivotal role in host defense through colonization resistance and development of the mucosal immune system; thus, the balance of commensal microbiota has an impact on health and the state of disease. AMPs have both antimicrobial and immunomodulatory properties, which directly and indirectly affect the composition of commensal microbiota. A recent report showed that intestinal commensal species increase resistance to antimicrobial activities of AMPs by up to four times by reducing the overall negative charge on cell surfaces via LPS modification [92]. Moreover, it has been reported that alterations in α-defensin expression can have a substantial impact on microbiota composition and that it manifests in association with changes in the IL-17A+CD4+ T cell count [93]. There is ongoing research on the association between vaginal dysbiosis and PTB. Although there may be variations across races, it is well known that Lactobacillus spp. (esp. Lactobacillus crispatus) dominance is associated with term birth, whereas Lactobacillus depletion, high diversity compositions, and the presence of bacterial vaginosis (BV)-associated bacteria increase the risk of PTB [94,95,96,97]. According to a recent study that investigated cervicovaginal microbiota and AMP expression, the risk of PTB can be lowered even with Mobiluncus curtsii or M. mulieris if Lactobacillus spp. relative abundance tertiles are present, but the risk of PTB is high even in the presence of Lactobacillus-spp.-dominated cervicovaginal microbiota if the β-defensin expression is low [39]. These results suggest that not only microbiome composition, but also innate immunity, including AMP action, can have a complex effect on the development of PTB. However, research on the role of AMP-microbiota interaction in PTB is still in its infancy, and further studies are needed.

## 3. Expression of AMPs in Pregnancy

AMPs, which are distributed throughout the female reproductive tract during pregnancy, play a role in preventing infection through their antimicrobial activities and modulating immune responses. HNP 1–3 were reported to be expressed in the vernix caseosa, chorion, placental trophoblasts, and amniotic fluid [41,98,99,100,101,102]. The levels of HNP 1–3 in the amniotic fluid are not markedly altered during pregnancy, but significantly increase during normal term parturition triggered by spontaneous labor [41]. β-defensin is widely expressed in pregnant uteri. HBD1–3 were reported to be expressed in the placenta and chorion trophoblast, amnion epithelium, and decidua, while HBD-1 and HBD-2 mRNA were expressed in the chorion, villus, and placental tissues [101,102,103,104]. The concentration of HBD-2 was not altered during pregnancy, whereas HBD-1 concentration was significantly higher in mid-pregnancy than in term [105,106]. High cathelicidin expression was observed in the fetal skin and vernix caseosa and within the amniotic fluid, with no marked changes in expression throughout pregnancy [107,108]. Lactoferrin expression has been reported in the amniotic fluid, amnion, cervix, mucus plug, and placenta [32,109]. Intra-amniotic lactoferrin concentration increased throughout pregnancy, and the lactoferrin concentration decreased in the amniotic fluid, whereas it increased in the umbilical cord plasma during spontaneous labor at term parturition. In addition, intra-amniotic infection is associated with an increased level of lactoferrin in the amniotic fluid [109,110].

## 4. AMPs Associated with PTB

In various clinical presentations related to PTB, such as preterm labor, PPROM, and cervical insufficiency, alterations in the expression of AMPs and their association with PTB have been reported. The lower female genital tract generally harbors *Lactobacillus*-dominant commensal bacteria, which prevent the attachment and invasion of microbial pathogens [94,111]. Recent metagenomic studies have identified a unique microbiome in the upper female reproductive tract, including the placenta, which was previously considered sterile [112]. The cervicovaginal microbial community is classified into six community state types (CSTs), four of which are dominated by a *Lactobacillus* spp. (*Lactobacillus crispatus* (CST I), *Lactobacillus gasseri* (CST II), *Lactobacillus iners* (CST III), or *Lactobacillus jensenii* (CST V)). The other two (CST IV-A and CST IV-B) lack a substantial number *of Lactobacillus* spp. and comprise a diverse array of anaerobic bacteria [113,114]. A healthy pregnancy is characterized by a shift to a less diverse and more *Lactobacillus* dominant CST [113]. Recent studies have shown that *Lactobacillus iners*-dominant vaginal communities (CST III) are associated with PTB, and vaginal dysbiosis characterized by lower levels of *Lactobacillus* spp. and high species diversity, as in BV, has been associated with increased risk of PPROM, PTB, and histologic chorioamnionitis [39,94,115,116]. BV and other reproductive tract infections are common during pregnancy, but PTB occurs only in certain subgroups, implying that PTB is affected by various factors, such as host defense mechanisms, in addition to the microbial pathogen. Balu et al. reported that women with intermediate BV at 24–29 weeks’ gestation were more likely to have higher vaginal fluid neutrophil defensin concentration, and women with elevated vaginal fluid neutrophil defensin concentration during mid-pregnancy had an increased risk for delivery before 32 weeks. However, elevated vaginal fluid neutrophil defensin concentration was not associated with PTB before 37 weeks [117]. Further, BV at <16 weeks’ gestation was associated with lower vaginal β-defensin 3 concentrations, but not HBD-2 or HNP 1–3 in African American-majority participants [118]. The association of HNP 3 levels and BV with PTB differed by race group: high vaginal HNP 1–3 levels at mid-pregnancy were associated with PTB in African American women, but vaginal HNP 1–3 levels were not related to PTB in non-Hispanic Whites [119]. In addition, a recent study reported that concentrations of cervicovaginal fluid cathelicidin and human neutrophil elastase at 10–24 weeks’ gestation were increased in women with cervical shortening and were predictive of PTB before 37 weeks, whereas another study demonstrated that higher vaginal β-defensin 2 levels were associated with a lower risk of PTB [39,120]. Moreover, in this study, even in *Lactobacillus* spp.-dominant cervicovaginal microbiota, low β-defensin 2 levels were associated with a higher risk of PTB [39]. In addition, a study on the association of maternal stress during pregnancy and PTB found that high stress was related to low cervicovaginal β-defensin 2 levels, and high stress and low cervicovaginal β-defensin 2 levels were risk factors for PTB [121]. The results of studies on alterations in the vaginal fluid AMP expression according to BV or the association between the AMP expression and the risk of PTB are inconsistent, which is attributable to differences in the type of AMPs, gestational weeks of sample collection, clinical features related to PTB, and ethnicity.

The HNP1–3 concentration in the amniotic fluid significantly increased in women with preterm labor with intra-amniotic infection, and HNP1–3 expression was markedly upregulated in women with PPROM [41]. In addition, the concentration of amniotic fluid HNP1–3 increased during both term parturition and PTB, and high concentration of amniotic fluid HNP1–3 was associated with intra-amniotic inflammation and histological chorioamnionitis in women with preterm labor, leading to PTB [41]. Moreover, amniotic fluid HNP1–3 levels increased markedly in women with subclinical intrauterine infection and exponentially according to the severity of histologic chorioamnionitis [98]. Several recent studies have revealed amniotic fluid β-defensin expression during normal pregnancy and changes in expression when intra-amniotic infection/inflammation or PPROM occurs. HBD-1 expression in the amniotic fluid is higher in the mid-trimester than in term, and HBD-1 expression is increased in women with intra-amniotic inflammation compared to those without intra-amniotic inflammation [106]. The HBD-2 concentration in the amniotic fluid did not change throughout the pregnancy and was markedly increased in women with intra-amniotic infection. Patients with preterm labor who delivered preterm had higher amniotic fluid HBD-2 concentrations than those who delivered at term. Further, HBD-2 expression was higher in women with intra-amniotic inflammation than in those without intra-amniotic inflammation [105]. Similar to HBD-2, the intra-amniotic concentration of HBD-3 also did not change throughout pregnancy, and women with spontaneous labor at term showed higher HBD-3 expression than those without labor. Moreover, HBD-3 levels were higher in women with intra-amniotic infection who delivered preterm due to preterm labor than in those who did not have an intra-amniotic infection. In women with PPROM, amniotic fluid HBD-3 concentrations were higher in women with PPROM and intra-amniotic infection than in those without intra-amniotic infection [40]. In addition, it was reported that cathelicidin may be used as a candidate marker to identify the presence of intra-amniotic infection in women with PPROM, since the level of cathelicidin in the amniotic fluid increases in the presence of intra-amniotic infection in PPROM patients [122]. In an experimental study using amnion epithelial cells, treating cells with LPS led to a marked upregulation of HBD-3 mRNA expression, and the HBD-3 protein in the amnion sections was intensively positive in women with histological chorioamnionitis who delivered preterm compared to the control patients who delivered at term [123]. Moreover, exposure of the amniotic membrane to IL-1β leads to increased secretion of AMPs, including HBD-2, HBD-3, cathelicidin, and elafin [124,125]. These results indicate that amnion epithelial cells can produce defensins in the amniotic fluid in response to infection or inflammatory stimuli and contribute to the innate immunity of the intra-amniotic cavity. The previously described alterations in the expression of AMPs associated with PTB are summarized in Table 1.

In fetal membranes, α-defensin 1 mRNA expression was markedly upregulated in the presence of histologic chorioamnionitis, and among those with histologic chorioamnionitis, α-defensin 1 mRNA expression was significantly higher in those with preterm labor with intact membranes than in those with PPROM [126]. In addition, HBD1–3 and elafin were present in the placenta and fetal membranes, and HBD-2 and elafin mRNA expressions were increased by proinflammatory cytokines in primary trophoblast cells [127]. These results suggest that AMPs in the placenta and fetal membranes also act as an immunologic barrier against infection by regulating their expression during pregnancy.

A recent whole-exome sequencing study identified rare mutations in genes encoding antimicrobial peptides/proteins (β-defensin 1 [DEFB1] and mannose-binding lectin [MBL2]) that were more frequent in neonates born to pregnancies complicated by PPROM [128,129]. Moreover, these genes have been previously linked to inflammatory bowel diseases and periodontal diseases. These inflammatory conditions have been reported to be associated with PTB, and alterations in AMP expression have been found to play an important role in the development of these inflammatory diseases. Therefore, these results suggest that mutations and damaging missense variants in the innate immunity or host defense genes may be associated with an increased risk of PTB. In addition, a recent RNA sequencing analysis revealed that women with cervical dilation at mid-pregnancy have markedly higher α-defensin 3 (DEFA3) gene and α-defensin 3 protein expression in blood compared to the normal group [130]. These findings suggest that α-defensin plays an important role in preventing microbial infection and inflammation in cervical insufficiency, where the mechanical and immunologic barrier of the cervix is disrupted. Moreover, among women with cervical dilation, downregulated DEFA3 gene expression at mid-pregnancy was prospectively associated with PTB. These findings seemed to contradict the results of previous studies that intra-amniotic inflammation/infection and preterm labor are associated with high defensin levels. However, considering that defensin deficiency is involved in the development of Crohn’s disease, and defensin functions as an immune response modulator through both pro-and anti-inflammatory properties, it is suggested that the differential expression of defensins, according to the severity of the infection and inflammatory response, can affect the pregnancy prognosis. Moreover, this tendency has also been observed in other recent studies. Low levels of β-defensin 2 were associated with an increased risk of PTB, and high maternal stress lowered β-defensin 2 levels, each of which increased the risk of PTB [39,121]. Taken together, these results suggested that the expression of AMPs increases when an inflammatory response such as intra-amniotic infection, PPROM, or cervical dilation occurs, but the expression of AMPs before the onset of symptoms may prospectively affect the development and progression of preterm parturition.

## 5. Conclusions and Future Perspectives

Several studies on significant alterations in the expression of AMPs in PTB-associated clinical presentations and correlation of the level of AMP expression with pregnancy prognosis suggest that AMPs play an important role in the development of PTB.

In view of the complex host defense properties of AMPs, such as antimicrobial and immunomodulatory properties and shaping of the composition of the commensal microbiota, many AMP-based therapeutic agents have recently been investigated, not only as new anti-infectives that can overcome the limitations of antibiotics related to drug-resistant pathogens, but also as immunomodulatory agents in a variety of indications, such as chronic inflammatory disorders and wound healing. As infections and microbial dysbiosis emerge as common causes of PTB, further research on functions and mechanisms of action of AMPs may enhance our understanding of the pathogenesis of PTB and provide promising treatment options.

## Figures and Tables

**Table 1 ijms-22-08905-t001:** Summary of the expression of AMPs associated with preterm birth.

AMPs	Gene	Site of Expression	Gestational Weeks of Sample Collection	Expression Associated with PTB	References
HNP 1-3	DEFA1,3	Vagina	24–29	PTB <32 weeks ↑	[117]
				PTB <37 weeks ↔	
			Mid-pregnancy	PTB ↑ in African American	[119]
				PTB ↔ in non-Hispanic Whites	
		Amniotic fluid	19–33	Intra-amniotic infection ↑	[41]
				PPROM ↑	
				Preterm parturition ↑	
			24–34	Subclinical intrauterine infection ↑	[98]
		Fetal membranes	23–33	Histologic chorioamnionitis ↑	[126]
				PPROM ↑	
Beta-defensins					
HBD-1	DEFB1	Amniotic fluid	20–32	IAI ↑	[106]
HBD-2	DEFB2	Vagina	16–28	PTB < 37 weeks ↓	[39]
		Amniotic fluid	20–32	Intra-amniotic infection ↑	[105]
				IAI ↑	
				Preterm delivery ↑	
HBD-3	DEFB3	Amniotic fluid	20–32	Intra-amniotic infection ↑	[40]
Cathelicidin	CAMP	Vagina	10–24	Cervical shortening ↑	[120]
				PTB < 37 weeks ↑	
		Amniotic fluid	24–34	PPROM with Intra-amniotic infection ↑	[122]

Abbreviations: HNP, human neutrophil peptides; PTB, preterm birth; PPROM, preterm premature rupture of membranes; HBD, human β-defensin; IAI, intra-amniotic inflammation; ↑, increased; ↔, not changed; ↓, decreased.

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
