# Peer review of "The Role of Antimicrobial Peptides in Preterm Birth"

_ijms, 2021, doi:10.3390/ijms22168905_

Round 1

Reviewer 1 Report

The article is well written and is beneficial for the readers. It debates AMPs' role in normal pregnancy course. 

Among various naturally spread anti-infective agents that have been so far discovered, AMPs are already not so new, but particularly important. These are versatile molecules, with highly specific antimicrobial activity, that are involved in human physiology.

I do have few points to address:

  1. Newer references are available on the topic and relevant for the article, but not included in the references. examples:

Ramuta TŽ, Šket T, Starčič Erjavec M, Kreft ME. Antimicrobial Activity of Human Fetal Membranes: From Biological Function to Clinical Use. Front Bioeng Biotechnol. 2021 May 31;9:691522. doi: 10.3389/fbioe.2021.691522. PMID: 34136474; PMCID: PMC8201995.

Al-Nasiry S, Ambrosino E, Schlaepfer M, Morré SA, Wieten L, Voncken JW, Spinelli M, Mueller M, Kramer BW. The Interplay Between Reproductive Tract Microbiota and Immunological System in Human Reproduction. Front Immunol. 2020 Mar 16;11:378. doi: 10.3389/fimmu.2020.00378. PMID: 32231664; PMCID: PMC7087453.

Tsonis O, Gkrozou F, Harrison E, Stefanidis K, Vrachnis N, Paschopoulos M. Female genital tract microbiota affecting the risk of preterm birth: What do we know so far? A review. Eur J Obstet Gynecol Reprod Biol. 2020 Feb;245:168-173. doi: 10.1016/j.ejogrb.2019.12.005. Epub 2019 Dec

Monin L, Whettlock EM, Male V. Immune responses in the human female reproductive tract. Immunology. 2020 Jun;160(2):106-115. doi: 10.1111/imm.13136. Epub 2019 Nov 11. PMID: 31630394; PMCID: PMC7218661.

Mei C, Yang W, Wei X, Wu K, Huang D. The Unique Microbiome and Innate Immunity During Pregnancy. Front Immunol. 2019 Dec 17;10:2886. doi: 10.3389/fimmu.2019.02886. PMID: 31921149; PMCID: PMC6929482.

2. As the conclusion section is a bit too long, like a continuance of the discussions section, I suggest only 3-4 lines of actual conclusions, as for the rest of the information, that announces the future therapeutic role, a new subsection would suffice. It is long known that AMPs are essential components of the innate immune system of vertebrates and the nonspecific host defense system of plants, fungi, and invertebrates that evolved over billion years ago as potent anti-infective agents against different viruses, bacteria, fungi, and parasites. There is abundant literature to introduce and debate the  AMPs therapeutic role in the article. example:

  1. Jonathan D Steckbeck, Berthony Deslouches & Ronald C Montelaro (2014) Antimicrobial peptides: new drugs for bad bugs?, Expert Opinion on Biological Therapy, 14:1, 11-14, DOI: 10.1517/14712598.2013.844227   Madanchi, H., Ebrahimi Kiasari, R., Seyed Mousavi, S.J. et al. Design and Synthesis of Lipopolysaccharide-Binding Antimicrobial Peptides Based on Truncated Rabbit and Human CAP18 Peptides and Evaluation of Their Action Mechanism. Probiotics & Antimicro. Prot. 12, 1582–1593 (2020). https://doi.org/10.1007/s12602-020-09648-5   da Cunha NB, Cobacho NB, Viana JFC, Lima LA, Sampaio KBO, Dohms SSM, Ferreira ACR, de la Fuente-Núñez C, Costa FF, Franco OL, Dias SC. The next generation of antimicrobial peptides (AMPs) as molecular therapeutic tools for the treatment of diseases with social and economic impacts. Drug Discov Today. 2017 Feb;22(2):234-248. doi: 10.1016/j.drudis.2016.10.017. Epub 2016 Nov 23. PMID: 27890668; PMCID: PMC7185764.   Giuliani, Andrea, Pirri, Giovanna and Nicoletto, Silvia. "Antimicrobial peptides: an overview of a promising class of therapeutics" Open Life Sciences, vol. 2, no. 1, 2007, pp. 1-33. https://doi.org/10.2478/s11535-007-0010-5      

Author Response

We appreciate your valuable comments for the revision of this manuscript.

  1. Newer references are available on the topic and relevant for the article, but not included in the references.

→ Thank you for your helpful comments. The references you mentioned have been added to the relevant parts of the manuscript. These references are very useful for the content of this manuscript and will be very helpful for further research.

  1. As the conclusion section is a bit too long, like a continuance of the discussions section, I suggest only 3-4 lines of actual conclusions, as for the rest of the information, that announces the future therapeutic role, a new subsection would suffice. It is long known that AMPs are essential components of the innate immune system of vertebrates and the nonspecific host defense system of plants, fungi, and invertebrates that evolved over billion years ago as potent anti-infective agents against different viruses, bacteria, fungi, and parasites. There is abundant literature to introduce and debate the AMPs therapeutic role in the article. example:
  1. Jonathan D Steckbeck, Berthony Deslouches & Ronald C Montelaro (2014) Antimicrobial peptides: new drugs for bad bugs?, Expert Opinion on Biological Therapy, 14:1, 11-14, DOI: 10.1517/14712598.2013.844227   Madanchi, H., Ebrahimi Kiasari, R., Seyed Mousavi, S.J. et al. Design and Synthesis of Lipopolysaccharide-Binding Antimicrobial Peptides Based on Truncated Rabbit and Human CAP18 Peptides and Evaluation of Their Action Mechanism. Probiotics & Antimicro. Prot. 12, 1582–1593 (2020). https://doi.org/10.1007/s12602-020-09648-5   da Cunha NB, Cobacho NB, Viana JFC, Lima LA, Sampaio KBO, Dohms SSM, Ferreira ACR, de la Fuente-Núñez C, Costa FF, Franco OL, Dias SC. The next generation of antimicrobial peptides (AMPs) as molecular therapeutic tools for the treatment of diseases with social and economic impacts. Drug Discov Today. 2017 Feb;22(2):234-248. doi: 10.1016/j.drudis.2016.10.017. Epub 2016 Nov 23. PMID: 27890668; PMCID: PMC7185764.   Giuliani, Andrea, Pirri, Giovanna and Nicoletto, Silvia. "Antimicrobial peptides: an overview of a promising class of therapeutics" Open Life Sciences, vol. 2, no. 1, 2007, pp. 1-33. https://doi.org/10.2478/s11535-007-0010-5      

→ Thank you for your helpful comment. We have deleted the following sentences from the conclusion section: “AMPs are present throughout the reproductive tract, have a wide range of antimicrobial properties to prevent infection, and play an important role in the maintenance of immune homeostasis by modulating immune responses. Moreover, the role of AMPs in shaping the composition of the commensal microbiota through AMP-microbiota interactions has been attracting attention.” We have also revised the conclusion section and have described the future therapeutic roles in more detail in a new subsection.

Reviewer 2 Report

The manuscript presented for evaluation is a review of the literature on antimicrobial peptides (AMPs) in the pathogenesis of preterm labor. The discussed topic is very interesting, also for medical practitioners. Extending the diagnostic possibilities in predicting preterm labor is extremely important. Knowing the risk gives you a chance for treatment. Antimicrobial peptides (AMPs) are another biochemical parameter investigated in the context of preterm labor. The results of the studies are promising, but further studies in large groups of patients are required to confirm the clinical value of the tests.
The manuscript is prepared carefully, with all the rules for the review. I propose to accept it for printing.

Author Response

We are grateful to the reviewer for the comments and suggestions, which have helped us to improve our manuscript. 

Round 2

Reviewer 1 Report

The article provides valuable information on a topic currently insufficiently known.  A separate chapter on the AMPs therapeutic role would have been beneficial.